# Clinical, Psychosocial, and Structural Factors Associated with the Detection of HIV Drug Resistance in Children Living with HIV in Kisumu, Kenya: Secondary Analysis of Data from the Opt4Kids Study

**DOI:** 10.3390/v17091246

**Published:** 2025-09-16

**Authors:** Andrea J. Scallon, Pooja Maheria, Patrick Oyaro, Katherine K. Thomas, Bhavna H. Chohan, Francesca Odhiambo, Evelyn Brown, Edwin Ochomo, Enericah Karauki, Nashon Yongo, Shukri A. Hassan, Marley D. Bishop, Ingrid A. Beck, Ceejay Boyce, Lisa M. Frenkel, Lisa Abuogi, Rena C. Patel

**Affiliations:** 1Elson S. Floyd College of Medicine, Washington State University, Spokane, WA 99202, USA; 2Division of Infectious Diseases, Department of Medicine, University of Alabama at Birmingham, Birmingham, AL 35233, USA; 3Health Innovations Kenya (HIK), Kisumu 40100, Kenya; 4LVCT Health, Nairobi 00202, Kenya; 5Department of Global Health, University of Washington, Seattle, WA 98105, USA; 6Centre for Microbiology Research, Research Care and Training Program, Kenya Medical Research Institute, Nairobi 00200, Kenya; 7UW Kenya, Nairobi 00200, Kenya; 8Department of Medicine, University of Washington, Seattle, WA 98105, USA; 9Center for Global Infectious Disease Research, Seattle Children’s Research Institute, Seattle, WA 98101, USA; 10Department of Pediatrics, University of Washington, Seattle, WA 98105, USA; 11Department of Laboratory Medicine and Pathology, University of Washington, Seattle, WA 98195, USA; 12Department of Pediatrics, University of Colorado, Denver, CO 80045, USA

**Keywords:** HIV, drug resistance mutation (DRM), children, viral failure, Kenya

## Abstract

Background: HIV drug resistance (DR) mutations can compromise antiretroviral therapy (ART) success among children living with HIV (CLHIV). We conducted a secondary analysis using data from a randomized control trial for ART monitoring among CLHIV in Kisumu County, Kenya from 2019 to 2023, to assess clinical, psychosocial, and structural factors associated with HIV DR. Methods: 704 CLHIV were followed for 12+ months, with characteristics captured at enrollment and follow-up visits in the “parent” randomized-controlled-trial (of point-of-care plasma viral load testing and for viremias ≥ 1000 copies/mL HIV genotyping for DR vs. standard-of-care) and an observational “extension” substudy (of participants on a dolutegravir-containing ART with genotyping performed on viremic specimens ≥ 200 copies/mL). A multivariate modified Poisson regression model was used to analyze factors associated with sequences yielding a Stanford HIVDR database DR penalty score (DR-PS) ≥ 30 to a nucleos(t)ides and/or non-nucleoside reverse transcriptase inhibitor, protease inhibitor (PI), and/or integrase inhibitor (INSTI). Results: Among 113 (16.1%) participants who underwent genotyping, 93 (82.3%) had a DR-PS ≥ 30. DR-PS ≥ 30 were associated with age 1–5 years (adjusted risk ratio (ARR) = 1.84; 95% confidence interval (CI): 1.07, 3.14), history of viremia ≥ 1000 copies/mL (ARR = 4.18; 95% CI: 2.77, 6.31), prescription of a PI- (ARR = 6.05; 95% CI: 3.43, 10.68) or INSTI-containing regimen (ARR = 1.83; 95% CI: 1.08, 3.11), poor adherence to ART (ARR = 1.91; 95% CI: 1.32, 2.76), lack of caregiver confidence in ART administration (ARR = 1.89; 95% CI: 1.11, 3.22), and mid-sized clinic populations (ARR = 0.55; 95% CI: 0.33, 0.92). Conclusion: Addressing social factors associated with DR-PS ≥ 30 may improve ART success among CLHIV.

## 1. Introduction

Pediatric HIV remains a serious global public health issue, with 2.4 million children living with HIV (CLHIV) < 19 years of age worldwide in 2024 [1]. Only 59% of CLHIV globally have HIV replication suppressed by antiretroviral therapy (ART), underscoring crucial and life-threatening gaps in optimal care for CLHIV [1,2]. African regions remain disproportionately impacted by HIV, accounting for >80% of the HIV burden among CLHIV [1]. Within the East African region, Kenya has approximately 62,000 CLHIV < 14 years of age and 4300 new infections among children annually [3]. Despite major advancements in viral load (VL) monitoring, healthcare delivery, and ART regimen availability including use of dolutegravir, ART coverage of CLHIV < 14 years of age in Kenya remains lower, at 70%, than their adult counterparts, with many children not achieving viral suppression (VS) [3,4,5]. Several high-HIV burden African countries, including Kenya, have signed the Dar es Salaam Declaration to commit to eliminating AIDS in children by 2030, yet critical challenges to optimizing VS and eliminating pediatric HIV transmission remain, including HIV drug resistance (DR) [6].

The emergence of HIV DR is a substantial threat to the elimination of HIV and the durability of ART, especially for CLHIV who are currently expected to receive life-long therapy. Numerous studies across several African regions have reported alarmingly high prevalence of DR among infants and children, including nearly 85% among CLHIV experiencing low-level viremia in Cameroon, 76% among non-virologically suppressed CLHIV < 19 years old living in Kenya, and up to 50% of infants < 18 months living with HIV in Nigeria [7,8,9]. In our parent study, conducted in Kenya, called “Optimizing VL suppression in Kenyan children on ART (Opt4Kids)”, 100% of children experiencing virologic failure (VF), defined as at least one HIV VL ≥ 1000 copies/mL, had at least one DR mutation (DRM) and 85% had at least one major DRM [10]. More recently, dolutegravir, an integrase strand transfer inhibitor (INSTI), has emerged as a well-tolerated, safe, and effective treatment for CLHIV, exhibiting easier dosing and a relatively high barrier to DR compared to other ART regimens for CLHIV [11,12,13,14]. With dolutegravir use rising globally, there are also increasing reports of DR among children on dolutegravir-containing ART [15,16,17,18]. Especially concerning in low and middle income countries (LMICs), such as Kenya, where availability of ART options are limited and obstacles to diagnostic testing and treatment are abundant, if DR to HIV is not urgently addressed, we risk a rise in HIV strains that are not responsive to any currently available ART and subsequent increases in HIV-related morbidity and mortality [19,20].

The factors driving DR among children in LMICs are numerous, diverse, and closely related to VS, and intertwining individual, social, and systemic influences lay the groundwork for DR acquisition among CLHIV. These factors include, but are not limited to, availability of child-appropriate ART formulations, fluctuations in body weight among children requiring dosing changes, lack of HIV diagnosis disclosure among children, low ART-adherence, perceived low-risk by clinic managers of patients acquiring DR, financial strains, and HIV-related stigma [8,21,22,23]. To enable effective ART and minimize the emergence of DR, an understanding of the individual-, household-, and societal-level factors that increase one’s risk for acquiring DR is needed. These insights may help identify subpopulations at greater risk for DR acquisition, determine which patients may be prioritized for treatment changes, and explore which interventions may be most effective to prevent DR. To better define factors associated with DR, we expand on our previous analysis of the Opt4Kids study [10] by assessing additional predictor variables and by including data from an “extension” substudy, or subset of the parent study, of children viremic during dolutegravir-containing ART.

## 2. Materials and Methods

### 2.1. Study Design and Setting

This secondary analysis used data from the “parent” Opt4Kids study [10], an open-label randomized control trial examining the impact of point-of-care (POC) VL and targeted DRM testing and clinical decision support among CLHIV on ART, as well as an “extension” substudy among a subset of the parent study participants. The parent study protocol and primary results from the Opt4Kids study have been detailed elsewhere [24,25]. Largely individual associations, including but not limited to child’s age, sex, ART regimen, and history of VF, with DRM among children enrolled in the parent Opt4Kids have been previously reported [10]. Here, we expand our variables to include 17 additional individual-, caregiver-, and household-level clinical, psychosocial, and structural factors. Additionally, we include HIV genotyping for DR from the parent and extension study, and genetic sequences are deposited in GenBank (accession numbers PX310287-PX310455 and NCBI BioProject accession number PRJNA1256474).

Briefly, the parent Opt4Kids study was conducted from 2019 to 2021 across five low-resource, high-HIV burden, high-volume public sector facilities in Kisumu County, Kenya, a region facing one of the highest burdens of HIV in the country [26,27]. Participants were randomized 1:1, stratified by facility and age group (ages 1–9 and 10–14 years), to either the intervention or standard-of-care (SOC) arm, and followed for at least 12 months. The intervention group received POC VL testing every 3 months using GeneXpert^®^ technology, targeted DRM testing for VL ≥ 1000 copies/mL, and clinical decision support for providers (i.e., multidisciplinary case review for interpretation of patient DRM testing results) via a clinical management committee. For children in the SOC arm, providers followed national Kenyan Ministry of Health SOC protocols for management of VF among CLHIV, including SOC VL testing every 6 months, enhanced adherence counseling for CLHIV with VF, and limited DRM testing for cases reviewed and approved by the NyaWest HIV Technical Working Group. Specifically, SOC DRM testing was approved for patients who were experiencing VF on first-line protease inhibitor (PI)-containing, or on second- or third-line ART with continued VF. Participants in both intervention and SOC arms underwent POC VL and DRM testing 12 months post-enrollment.

From 2022 to 2023, we conducted the extension substudy to better understand DR to dolutegravir as part of a larger study [28]. CLHIV enrolled in the parent study were recruited from four of the five Opt4Kids facilities (the fifth one was excluded due to lowest enrollment numbers and furthest distance from the study team’s central location). Eligibility included being 1–14 years old, newly initiating or already receiving ART at one of the study facilities, having a VL ≥ 200 copies/mL within the last 6 months while on dolutegravir-containing ART, and having caregiver consent. Enrollees were followed every 6 months for a total of an additional 12 months. All samples collected in the extension substudy underwent POC VL testing on site and storage for genotyping at the end of the study. Together, we refer to the parent and extension studies as “Opt4Kids studies”.

Initially in 2019, first-line ART regimens for CLHIV < 14 years of age in Kenya included combinations of two nucleoside reverse transcriptase inhibitors (NRTIs), lamivudine with abacavir or zidovudine, with a non-nucleoside reverse transcriptase inhibitor (NNRTI) efavirenz (preferred for those >3 years of age) or the PI lopinavir/ritonavir (preferred for those <3 years of age). Starting in 2020, dolutegravir was added as the preferred ART regimen for those weighing >35 kg [29], and in 2022, it became first-line treatment for those weighing ≥20 kg [30,31]. Optimal second- and third-line ART regimens for CLHIV vary and are determined by genotyping. Second-line ART regimens may include dolutegravir and lamivudine with abacavir or zidovudine, and possible third-line regimens include combinations of darunavir/ritonavir, lamivudine, dolutegravir or etravirine, and abacavir or zidovudine [31].

Opt4Kids was ethically and scientifically approved by institutional review boards at AMREF (Kenya) and the University of Washington and University of Colorado (United States), Kenya National Commission for Science, Technology, and Innovation, and the Kenya Pharmacy and Positions Board.

### 2.2. Participants and Sample Size

For this secondary analysis, we included data from all 704 children participating in both the parent and extension Opt4Kids studies. Details on participant exclusion criteria for the parent study are detailed elsewhere [25]. No power or sample size calculations were conducted for this secondary analysis.

### 2.3. Variables

#### 2.3.1. Primary Analytic Outcome

The primary analytic outcome is detection of HIV DR to NRTI, NNRTI, PI, or INSTI drug classes at any point during the Opt4Kids studies with a DR penalty score (DR-PS) ≥ 30 to any antiretroviral (ARV) agent as assessed by the Stanford HIV DR database (HIVdb) [32] version 9.8 (2025-01-05) accessed on 21 April 2025. For children with ≥ 2 DR genotypes, the specimen with the greatest DR-PS was analyzed. Sanger sequences in the parent study were generated from viremic specimens with Applied Biosystems HIV-1 Genotyping kit using an Applied Biosystems 3130xl Genetic Analyzers (ThermoFisher Scientific, Nairobi, Kenya) at the KEMRI-CDC HIV Research and Sanger 3730xl at the Kenya National HIV Reference Laboratories [33,34] analyzed for DR to NRTI, NNRTI and PI. Sequences from the extension study were derived from viremic specimens while on dolutegravir-containing ART by laboratory-developed methods using PacBio platform or Sanger sequencing (passing NIH Virology Quality Assurance Program) [28] in Seattle and analyzed for DR to NRTI, NNRTI, PI and INSTI. All sequence analysis used the Stanford HIVdb [32] version 9.8 (2025-01-05) accessed on 21 April 2025 which assesses together all DRMs affecting each ARV to generate the DR-PS. Despite a DRM penalty score of ≥15 predicting low-level DR [35], there is some consensus that a score of ≥ 30 is a more clinically significant cutoff to measure DR [36]. The International Antiviral Society-USA has defined “major” and “minor” mutations for all drug classes based on expert consensus [37] which change over time, and because these change over time, we used the DR-PS cutoff approach to make our results more easily reproduceable. In our analysis, participants with DR-PS < 30 are grouped together with participants who did not undergo any DR testing.

Sensitivity analyses that use DR-PS of ≥15 or ≥60 are included in the Appendix A.

#### 2.3.2. Associated Risk Factors

We used each child’s clinical, caregiver’s clinical, child’s psychosocial, caregiver’s psychosocial, household psychosocial, and structural characteristics collected in the Opt4Kids studies, as detailed below. Some variables were captured only at enrollment (e.g., history of VF prior to study enrollment) while others were captured repeatedly, longitudinally until the end of study follow-up. For repeating measures captured longitudinally (e.g., caregiver intimate partner violence (IPV), stigma, depression), we interpreted and analyzed participants who had experienced a repeating measure at any point (i.e., answered “yes”), despite potentially changing responses during follow up, as having experienced the measure.

#### 2.3.3. Child and Caregiver Clinical Characteristics

We define clinical characteristics as the personal and clinical attributes that influence an individual’s health outcome. We include the following child clinical characteristics: age, sex, history of VF within two years prior to study, base drug in ART regimen at time of DR test analysis, and co-administered NRTIs at time of DR test analysis. Of note, for those participants who did not undergo DR testing, we developed a “proxy” time of DR test based on the median time from enrollment to DR test used in this analysis for those who did undergo DR testing. We measure the history of VF as any VL result ≥ 1000 copies/mL within two years prior to parent study enrollment. Clinical characteristics of the caregiver include age and self-reported VS status.

#### 2.3.4. Child Psychosocial Characteristics

We define psychosocial characteristics as the traits that arise from the impact of social factors, including interpersonal relationships, community norms, and cultural influences, on behavior and mental health [38]. The psychosocial child characteristics in our study include school level and type, ART adherence, and awareness of diagnosis. ART adherence is measured as self-report (or caregiver report) as per the national Kenyan guidelines, with adherence to ≥95% of medications, 85–94% of medications, and <85% of medications in the last 30 days being classified as “good”, “inadequate” and “poor”, respectively [31].

#### 2.3.5. Caregiver Psychosocial Characteristics

Caregiver psychosocial characteristics include caregiver marital status, educational attainment, stigma, depression, HIV literacy, medicine administration confidence, and experience of IPV. Caregiver stigma is measured via a modified version of the People Living with HIV (PLHIV) Stigma Index 2.0 survey in Kenya and converted into a binary variable (i.e., yes/no) based on responses of “agree” or “strongly agree” to survey questions regarding internalized, enacted, and anticipated stigma [39]. Caregiver depression is measured via the Patient Health Questionnaire 9, with a score of ≥ 10 indicating having depression [40]. We evaluate caregivers’ confidence in administering medications by using a de novo 4-point Likert scale developed by our team with responses that range from “not confident” to “very confident”. In our study, responses of “very confident” and “mostly confident” are categorized as “confident”, while responses of “somewhat confident” and “not confident” are categorized as “not confident”. Caregiver IPV, defined as physical or sexual violence, stalking, or other controlling or aggressive behaviors by an intimate partner [41], is measured via a modified version of the World Health Organization’s (WHO) Violence Against Women study instrument [42] and converted into a binary variable (i.e., yes/no) to represent any IPV experience throughout the study period.

#### 2.3.6. Household Psychosocial Characteristics

We define household-level characteristics as attributes of a group of people who regularly share resources, activities, and costs [43]. For household-level psychosocial characteristics, we include food insecurity, other children living in the home, and other CLHIV living in the home. Food insecurity is measured via the Household Food Insecurity Access Scale which includes categories of food secure, mildly food insecure, moderately food insecure, and severely food insecure [44]. Participants whose score corresponds to “mildly food insecure”, “moderately food insecure,” or “severely food insecure” are categorized as “food insecure,” while participants whose score corresponds to “food secure” are categorized as “food secure”.

#### 2.3.7. Structural Characteristics

We define structural characteristics as institutional, cultural, and policy-related factors that influence patterns of advantage and overall health [45], including rurality, travel time, and clinic volume. Rurality is defined as clinic location (i.e., urban, semi-urban, or rural) and travel time from their primary residence to the facility was self-reported by caregivers at enrollment. Clinic volume is classified as high, medium, or low based on HIV patient visits per month at the facility.

### 2.4. Data Sources and Management

Trained research staff used direct, electronic data entry to collect data via tablets onto standardized collection forms created in REDCap. Collected data included demographic, clinical (e.g., health indicators), and laboratory data, including VL and DR results, acquired from routine medical records, a national online VL database, and facility laboratory logbooks. Research staff also administered questionnaires, in-person or by phone, for self-reported data. Validated instruments were utilized in the questionnaires where available.

### 2.5. Statistical Methods

We performed descriptive analyses, utilizing frequencies and percentages. We report on the prevalence of HIV DR by specific ARV medication using a DR-PS ≥ 30 as defined by the Stanford HIVdb. We utilized a modified Poisson regression model with a log link and robust standard error estimations, as we have done in the primary outcomes analysis [25]. We report unadjusted and adjusted relative risk and 95% confidence intervals, with statistical significance set at *p* < 0.20 for potential predictor variables to be included from the univariate models in the multivariate model. Statistical analyses were conducted using Python version 3.11.5.

## 3. Results

### 3.1. Individual, Household, and Structural Characteristics of Participants

A total of 704 CLHIV were enrolled in the parent study with a median age of 10 years (interquartile range [IQR] 7, 12) (Figure 1; Table 1). Of the 704 CLHIV from the parent study, 74 were enrolled in the extension substudy. The participants’ ART regimens at the time of their greatest DR-PS or if no viremias, at time of their first DR test, include 363 (51.6%) on NNRTI-containing regimens, 285 (40.5%) on PI-containing regimens, 55 (7.8%) on INSTI-containing regimens, and 1 (0.1%) was on an NRTI-only regimen. The most common full ART regimens at this time were abacavir/lamivudine/efavirenz, abacavir/lamivudine/lopinarvir/ritonavir, and zidovudine/lamivudine/lopinavir/ritonavir. A history of VF within two years prior to enrolling into the parent study was reported for 144 (20.5%) participants. Primary caregivers were mostly over the age of 24 years (*n* = 661, 93.9%) and, among those living with HIV were virally suppressed (*n* = 431, 61.2%). Some caregivers experienced depression (*n* = 252, 35.8%), held HIV-related stigmatizing beliefs (*n* = 105, 14.9%), or experienced IPV (*n* = 163, 23.2%). Nearly all participant households experienced some food insecurity over the study period (*n* = 687, 97.6%). Participants largely attended clinics in urban (*n* = 421, 59.8%) or semi-urban areas (*n* = 158, 22.4%).

### 3.2. HIV Drug Resistance (DR) Testing and DR Penalty Scores (DR-PS) ≥ 30

Among the 704 CLHIV, 113 (16.1%) had 190 DR tests throughout the study periods, of which 93 (82.3%) had DR-PS ≥ 30 in one or more test results (Table 1). In comparison to the 591 (83.9%) CLHIV without testing combined with the 20 (2.8%) with a DR-PS < 30, those with DR-PS ≥ 30 were more likely to have a history of VF within two years prior to the parent study (14.2% vs. 61.3%), have poor adherence (18.3% vs. 36.6%) be on a PI- (39.3% vs. 48.4%) or INSTI-containing (4.4% vs. 30.1%) regimen at time of DR test analysis, and be age 1–5 years (9.0% vs. 25.8%).

Across the 93 participants with any DRM assessed by Stanford HIVdb, NRTI and NNRTI associated mutations were most prevalent and included M184I/MIV/MV/V (50%), L74V/I (13%), and Y115F/YF (13%), and K103N/KM/KN (28%), G190A/E/GA (19%), and Y181C/YC (11%), respectively (Figure 2b,c). PI (*n* = 6, 6.5%) or INSTI (*n* = 5, 5.4%) mutations were less common (Figure 2d,e). Importantly, over 70% of the 93 participants had DR conferring resistance to emtricitabine and lamivudine. Over 80% had DR to efavirenz (*n* = 82, 88.2%) and nevirapine (*n* = 82, 88.2%), with fewer conferring cross-resistance to rilpivirine (*n* = 36, 38.7%) (Figure 3). Few (<5%) of these participants’ sequences encoded DR to PI and INSTI. Most participants were resistant to multiple ARVs within a drug class. 82 (88.2%) participants had a DR-PS ≥ 30 to two or more NNRTIs, and 69 (74.2%) had a DR-PS ≥ 30 to two or more NRTIs (Table 1).

### 3.3. Associations with DR-PS ≥ 30

#### Individual, Household, and Structural Characteristics

Comparisons of participants’ risk factors between those with versus without DR-PS ≥ 30 found that DR-PS ≥ 30 was associated with children’s age of 1–5 years, history of VF within two years prior to study enrollment, PI- or INSTI-containing regimens at time of DR test analysis, self-reported poor adherence, and lack of caregiver drug administration confidence (Table 2; Figure 4). Receiving HIV care at a clinic with a midsized population was associated with a DR-PS < 30. A DR-PS ≥ 30 was not significantly associated with sex, ART-base, co-administered NRTIs, food insecurity, school level, school type, child awareness of status, multiple factors related to participants’ caregivers (i.e., marital status, age, VS, educational attainment, depression, stigma, HIV literacy, IPV), having other children or CLHIV in the home, travel time to the clinic, or clinic location.

## 4. Discussion

Given that CLHIV currently need life-long ART, there is an urgent need to address the underlying social causes of ART non-suppression and selection of HIV DR to ensure longevity of already-limited therapy regimens [46]. Our study confirms risk factors identified in our earlier analysis of Opt4Kids cohort [10] plus identifies novel risk factors associated with DR. We confirm that HIV DR (defined by a Stanford HIVdb DR-PS ≥ 30) is associated with a younger age (1–5 years) [10], a history of VF [10], and being prescribed a PI- [10] or INSTI-containing regimen at time of DR test analysis (the latter two likely being surrogate markers for HIV treatment experience). Newly identified caregiver, psychosocial, and structural factors, including self-reported poor adherence and a lack of caregiver confidence in ART administration were found to be associated with DR, and attending a clinic with a midsized population was found to be protective against having any DR. In this analysis, we also reveal that food insecurity, awareness of status, and caregiver factors (e.g., depression, VS, stigma, educational attainment, IPV, or HIV literacy) are not associated with HIV DR.

The detection of HIV DR to dolutegravir was low (<5%) across participants tested during viremic episodes. The paucity of DR to dolutegravir is likely due in part to relatively few people observed during dolutegravir-based ART and most achieving sustained suppression of HIV replication. However, several participants had high prevalence of resistance to multiple NRTIs, including lamivudine and tenofovir, which has the potential to render some children on functional dolutegravir monotherapy [28].

The significant associations identified in our analysis (e.g., base drug in ART regimen at time of DR test analysis, etc.) could potentially be driven by any specific HIV DRM, including but not limited to K103N or Y181C, although not directly assessed in our analysis. Prior studies have corroborated how individual-, household-, and societal-level clinical, psychosocial, and structural vulnerabilities impact VS and adherence, yet fewer studies have indicated how these factors influence DRM acquisition [47,48,49,50]. A study conducted among a group of children and adolescents in rural southern Tanzania indicated that younger age, female sex, and poor adherence were risk factors for acquisition of HIV DR, mostly consistent with our findings [51]. Conceptualizing the development of DR as solely due to biomedical or clinical factors at the individual level may be limiting. Thus, our current analysis, that moves beyond individual-level clinical factors to start examining caregiver- and household-level factors that intersect with psychosocial and structural elements, remains relevant. In our study, the relationship between individual factors with DR acquisition may exist due to wider, underlying social or structural reasons, including but not limited to poverty, stigma, gender inequality, and social norms [52,53]. For example, unequal access to economic resources and systemically reinforced norms, including fear of blame or ridicule by peers after HIV status disclosure and expectations of females in regard to sexual behavior, reduce females’ uptake of HIV testing, services, and treatment, potentially explaining why female sex in has shown to be a significant risk factor for DR acquisition among other studies [51,52,53,54]. When designing interventions to address the rise in DR, Campbell and Cornish’s conceptualization of social context offers a useful model, drawing on three dimensions that shape social environments, including material (e.g., resources, economic status), symbolic (e.g., meanings, ideologies, and norms), and relational (e.g., leadership and relationships with outside organizations, politicians, etc.) contexts [55]. Effectively addressing DR acquisition among girls and women, then, goes beyond overcoming individual-level challenges to HIV care and also calls for a shared commitment by local agencies and leadership in all levels of the government to provide tailored structural programs that confront the underlying norms of stigma, violence, and unequal access to economic resources that may contribute to DR acquisition [55,56,57,58,59]. Thus, comprehensively addressing the deeper social contexts and environments through which DR emerges and creating “health-enabling social environments”, or social environments that promote and support health-enhancing behavior, may help shape future interventions that tackle DR acquisition to better address the social root causes of this issue [55].

We anticipated that living farther away from a treatment facility, a potential barrier to maintaining consistent care, would be associated with DR; seemingly paradoxically, our analysis suggests that living farther from a facility, a structural factor, may serve as a protective factor to DR acquisition, although the relationship was not statistically significant. According to existing literature across various African regions, it is not uncommon for PLHIV to travel beyond their nearest facility to receive care, with one study in rural South Africa showing that PLHIV were nearly four times more likely to travel greater distances than patients seeking care for other health issues [60,61,62]. Travelling farther distances from home communities to seek care could provide privacy, or anonymity, indirectly promoting PLHIV to attend appointments and adhere to treatment while avoiding stigma, discrimination, and social exclusion due to inadvertent disclosure [61,62,63]. Yet, our study finds that caregiver stigma was not associated with acquisition of DR, suggesting alternative explanations for travelling farther to seek care may exist. Other studies have suggested that PLHIV may travel greater distances due to a lack of knowledge of nearby facilities, or to seek specialized HIV care at facilities with higher perceived quality of care, such as at tertiary hospitals [61,64]. While some people to travel greater distances to seek care, data from multiple studies find that transportation-related barriers, including distance, high transportation costs, poor road conditions, or lack of reliable transportation, are associated with decreased levels of ART adherence, increased rates of defaulting on ART, and reduced use of voluntary counselling and testing services across several African regions [65]. Additional research could better define intricacies of factors, such as stigma or improved perceived quality of care, that drive individuals to travel farther to seek HIV care in a manner that allows better treatment of these issues by caregivers.

While our quantitative analysis has strengths in understanding potential predictors of pediatric HIV DR and current challenges in addressing DR in Kenya, the study also has several limitations. First, our interpretations do not account for the timing or the circumstances through which DR was acquired among participants. For example, among children currently on INSTI-containing regimens, mutations may have been previously selected while receiving NNRTI- or PI-containing ART. We did not assess longitudinal data measuring when mutations were selected, so our analyses should be interpreted with an understanding that mutations may have simply persisted on prior regimens rather than being newly selected on current regimens. Secondly, a significant portion of data was self-reported by caregivers, rather than obtained via objective measures, so social desirability bias, etc., may exist. Third, our study lacks more measures of social or structural vulnerability to various HIV outcomes; a more comprehensive approach to understanding social contexts, including the material, symbolic, and relational contexts can better inform future work. Lastly, our analysis largely considers presence of DR as a proxy for viremia, largely due to non-adherence, and not DR necessitating a regimen change from the current one (our analysis of DR leading to changes in regimen and improvement in VS from the parent study are reported elsewhere [10]). Lastly, our parent study did not analyze sequence data encoding integrase, and while captured in our extension substudy, which found the prevalence of DR was low, relatively few children were followed for a short time while on dolutegravir-containing ART.

## 5. Conclusions

Our study emphasizes that multiple individual- and household-level clinical, psychosocial, and structural factors must be addressed to enable ART suppression of HIV replication and avoid selection of DR. In our cohort, assisting families with ART adherence and boosting caregiver confidence in ART administration could increase rates of ART suppression. Next steps could include development of interventions that identify individuals and their families at risk for DR acquisition or enhancing education and counseling at the clinic level. Additional research is needed to elucidate the pathways, especially the social contexts, through which these factors might influence and be modified for treatment success among CLHIV.

## Figures and Tables

**Figure 1 viruses-17-01246-f001:**
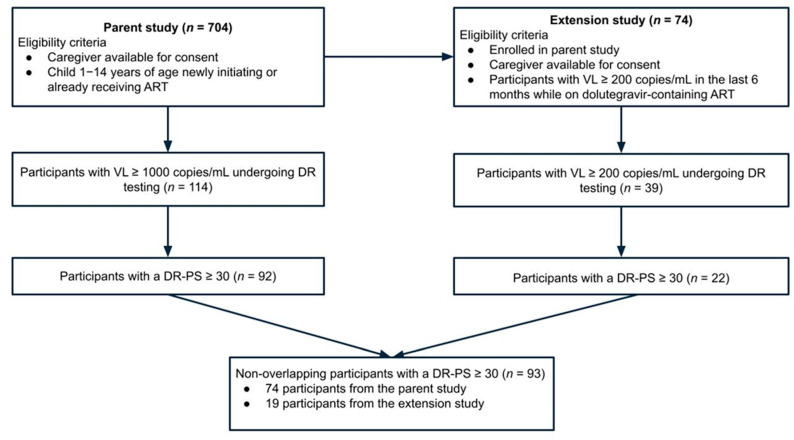
Flow diagram of participants included for analysis from Opt4Kids parent and extension studies (*n* = 704).

**Figure 2 viruses-17-01246-f002:**
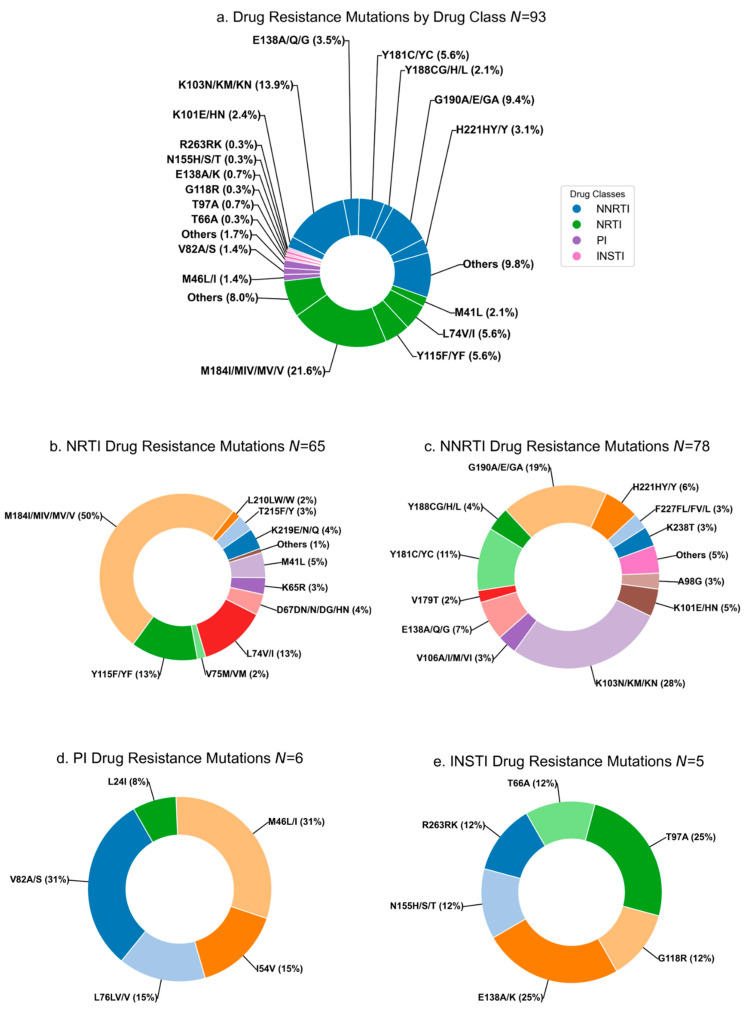
Distribution of HIV drug resistance mutations defined in Stanford HIVdb taken from sequence with greatest drug resistance penalty score *N* = number of children with mutations to the respective drug class(es). (**a**) Distribution of all drug resistance mutations by drug class and select codons; (**b**) distribution of NRTI mutations; (**c**) distribution of NNRTI mutations; (**d**) distribution of PI mutations; and (**e**) distribution of INSTI mutations. (NRTIs—nucleoside reverse transcriptase inhibitors; NNRTIs—non-nucleoside reverse transcriptase inhibitors; INSTI—integrase strand transfer inhibitors; PIs—protease inhibitors). Note: Mutations categorized as “Other” had a prevalence below the 1.3% cutoff threshold established by our team and were therefore excluded from the figure.

**Figure 3 viruses-17-01246-f003:**
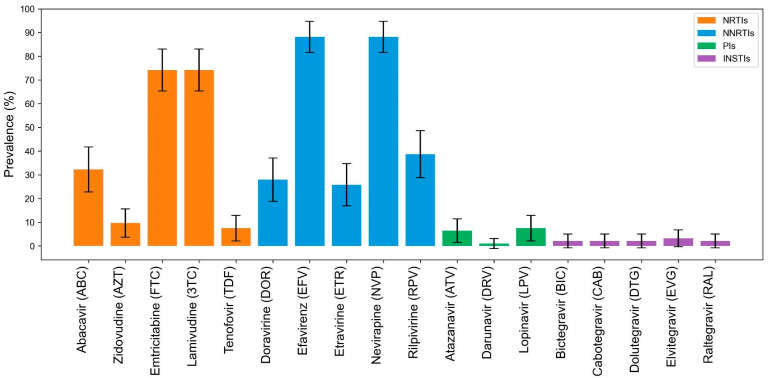
Prevalence (and 95% confidence intervals) of Stanford HIVdb drug resistance penalty scores ≥ 30 to available antiretroviral agents derived from mutations detected in participants Note: Abbreviations: NRTIs—nucleoside reverse transcriptase inhibitors; NNRTIs—non-nucleoside reverse transcriptase inhibitors; INSTI—integrase strand transfer inhibitors; PIs—protease inhibitors.

**Figure 4 viruses-17-01246-f004:**
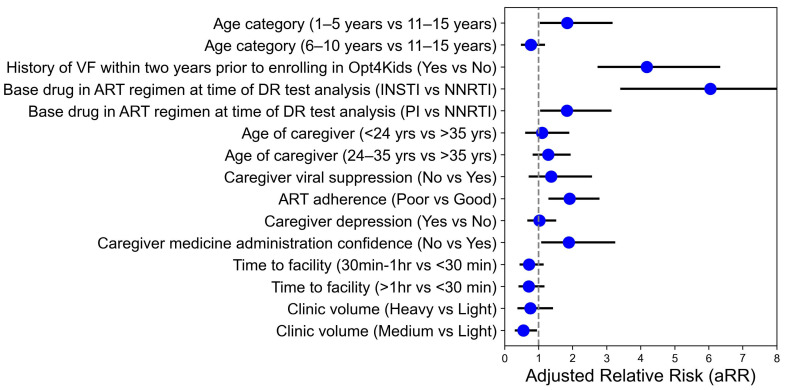
Forest plot of factors associated with a drug resistance penalty score ≥ 30 to any antiretroviral agent assessed by Stanford HIVdb detected in children enrolled in Opt4Kids parent and/or extension studies (*n* = 704) from the multivariate model output. Note: Abbreviations: ART—antiretroviral therapy; DRM—drug resistance mutation; NNRTI—non-nucleoside reverse transcriptase inhibitor; INSTI—integrase strand transfer inhibitor; PI—protease inhibitor.

**Table 1 viruses-17-01246-t001:** Characteristics of children enrolled in Opt4Kids by detection of HIV drug resistance (*n* = 704).

	Total *n* = 704	Children Without DR testing or DR-PS < 30 *n* = 611	Children with DR-PS ≥ 30 *n* = 93
Characteristics	*N* (%)	*N* (%)	*N* (%)
Child Clinical Characteristics			
Age, median (IQR)	10 (7, 12)	10 (7, 12)	9 (5, 12)
Age category (years)			
1–5	79 (11.2)	55 (9.0)	24 (25.8)
6–10	301 (42.8)	269 (44.0)	32 (34.4)
11–15	324 (46.0)	287 (47.0)	37 (39.8)
Sex			
Male	360 (51.1)	313 (51.2)	49 (48.04)
Female	344 (48.9)	298 (48.8)	53 (51.96)
History of VF within two years prior to enrolling in Opt4Kids			
Yes	144 (20.5)	87 (14.2)	57 (61.3)
No	504 (71.6)	476 (77.9)	28 (30.1)
Missing	56 (8.0)	48 (7.9)	8 (8.6)
Base drug in ART regimen at time of DR test analysis			
NNRTI-containing	363 (51.6)	343 (56.1)	20 (21.5)
PI-containing	285 (40.5)	240 (39.3)	45 (48.4)
INSTI-containing	55 (7.8)	27 (4.4)	28 (30.1)
Only NRTI-containing	1 (0.1)	1 (0.2)	0
Co-administered NRTIs at time of DR test analysis			
ABC + 3TC	479 (68.0)	425 (69.6)	54 (58.1)
AZT + 3TC	135 (19.2)	115 (18.8)	20 (21.5)
TDF + 3TC	90 (12.8)	71 (11.6)	19 (20.4)
Full ART regimen at the time of DR test analysis ^1^			
NNRTI-containing			
ABC + 3TC + EFV	203 (28.8)	194 (31.8)	9 (9.7)
ABC + 3TC + NVP	47 (6.7)	42 (6.9)	5 (5.4)
AZT + 3TC + NVP	41 (5.8)	40 (6.5)	1 (1.1)
AZT + 3TC + EFV	12 (1.7)	11 (1.8)	1 (1.1)
TDF + 3TC + EFV	60 (8.5)	56 (9.2)	4 (4.3)
PI-containing			
ABC + 3TC + LPV/r	198 (28.1)	172 (28.2)	26 (28.0)
AZT + 3TC + LPV/r	72 (10.2)	58 (9.5)	14 (15.1)
AZT + 3TC + ATV/r	6 (0.9)	4 (0.7)	2 (2.2)
TDF + 3TC + ATV/r	4 (0.6)	3 (0.5)	1 (1.1)
TDF + 3TC + LPV/r	4 (0.6)		
INSTI-containing		2 (0.3)	2 (2.2)
ABC + 3TC + DTG	31 (4.4)	17 (2.8)	14 (15.1)
AZT + 3TC + DTG	3 (0.4)	1 (0.2)	2 (2.2)
TDF + 3TC + DTG	21 (3.0)	9 (1.5)	12 (12.9)
Other			
TDF + 3TC + DTG + ATV/r	1 (0.1)	1 (0.2)	0
AZT + 3TC + ABC	1 (0.1)	1 (0.2)	0
Frequency of resistance to multiple drugs (DR-PS > 30) within an ART drug class ^2^			
NRTI—1 drug	-	-	3 (3.2)
NRTI—>2 drugs	-	-	69 (74.2)
NNRTI—1 drug	-	-	0
NNRTI—>2 drugs	-	-	82 (88.2)
PI—1 drug	-	-	1 (1.1)
PI—>2 drugs	-	-	6 (6.5)
INSTI—1 drug	-	-	0
INSTI—>2 drugs	-	-	3 (3.2)
Caregiver Clinical Characteristics			
Age of caregiver			
<24 years	40 (5.7)	31 (5.1)	9 (9.7)
24–35 years	325 (46.2)	272 (44.5)	53 (57.0)
>35 years	336 (47.7)	305 (49.9)	31 (33.3)
Missing	3 (0.4)	3 (0.5)	0
Caregiver viral suppression			
Yes	431 (61.2)	379 (62.0)	52 (55.9)
No	39 (5.5)	30 (4.9)	9 (9.7)
Not applicable	135 (19.2)	116 (19.0)	19 (20.4)
Unknown	99 (14.1)	86 (14.1)	13 (14.0)
Child Psychosocial Characteristics			
School level			
Nursery	124 (17.6)	107 (17.5)	17 (18.3)
Primary	512 (72.7)	454 (74.3)	58 (62.4)
Secondary	4 (0.6)	4 (0.7)	0
Missing	64 (9.1)	46 (7.5)	18 (19.4)
School type			
Day school	631 (89.6)	557 (91.2)	74 (79.6)
Boarding	4 (0.6)	4 (0.7)	0
Mix of day and boarding	7 (1.0)	6 (1.0)	1 (1.1)
Missing	62 (8.8)	44 (7.2)	18 (19.4)
ART adherence			
Good	558 (79.3)	499 (81.7)	59 (63.4)
Poor	146 (20.7)	112 (18.3)	34 (36.6)
Adolescence awareness of status			
No	408 (58.0)	352 (57.6)	56 (60.2)
Yes	296 (42.0)	259 (42.4)	37 (39.8)
Caregiver Psychosocial Characteristics			
Caregiver marital status			
Married	442 (62.8)	380 (62.2)	62 (66.7)
Unmarried	261 (97.1)	230 (37.6)	31 (33.3)
Unknown	1 (0.1)	1 (0.2)	0
Educational attainment of caregiver			
No Education	29 (4.1)	27 (4.4)	2 (2.2)
Primary	398 (56.5)	344 (56.3)	54 (58.1)
Secondary & above	277 (39.3)	240 (39.3)	37 (39.8)
Caregiver depression			
Yes	252 (35.8)	205 (34.75)	40 (43.0)
No	451 (64.1)	398 (65.1)	53 (57.0)
Missing	1 (0.14)	1 (0.2)	0
Caregiver stigma			
Yes	105 (14.9)	93 (15.2)	12 (12.9)
No	505 (71.7)	436 (71.4)	69 (74.2)
Missing	94 (13.4)	82 (13.4)	12 (12.9)
HIV literacy			
Yes	331 (47.0)	287 (47.0)	49 (52.7)
No	373 (53.0)	324 (53.0)	44 (47.3)
Medicine administration confidence			
Yes	665 (94.5)	582 (95.3)	83 (89.2)
No	39 (5.5)	29 (4.7)	10 (10.8)
Caregiver intimate partner violence			
Yes	163 (23.2)	140 (22.9)	23 (24.7)
No	540 (76.7)	470 (76.9)	70 (75.3)
Missing	1 (0.1)	1 (0.2)	0
Household Psychosocial Characteristics			
Food insecurity			
No	16 (2.3)	15 (2.5)	1 (1.1)
Yes	687 (97.6)	595 (97.4)	92 (98.9)
Missing	1 (0.1)	1 (0.2)	0
Other children in the home			
Yes	630 (89.5)	544 (89.0)	86 (92.5)
No	71 (10.1)	64 (10.5)	7 (7.5)
Missing	3 (0.4)	3 (0.5)	0
Other CLHIV in the home			
Yes	107 (15.2)	96 (15.7)	11 (11.8)
No	593 (84.2)	512 (83.8)	81 (87.1)
Missing	4 (0.6)	3 (0.5)	1 (1.1)
Structural Characteristics			
Travel time to facility			
<30 min	231 (32.8)	188 (30.8)	43 (46.2)
30 min to 1 h	309 (43.9)	278 (45.5)	31 (33.3)
>1 h	164 (23.3)	145 (23.7)	19 (20.4)
Clinic location			
Urban	421 (59.8)	367 (60.1)	54 (58.1)
Semi-urban	158 (22.4)	132 (21.6)	26 (28.0)
Rural	125 (17.8)	106 (17.97)	13 (14.0)
Clinic population volume			
Heavy	203 (28.8)	180 (29.5)	23 (24.7)
Medium	439 (62.4)	383 (62.7)	56 (60.2)
Light	62 (8.8)	48 (7.9)	14 (15.1)

Abbreviations: ABC—Abacavir; ART—antiretroviral therapy; ATV/r—Atazanavir/Ritonavr; AZT—Zidovudine; CLHIV—children living with human immunodeficiency virus; DR—drug resistance; DR-PS—drug resistance penalty score; DTG—Dolutegravir; EFV—Efavirenz; LPV/r—Lopinavir/Ritonavir; INSTI—integrase strand transfer inhibitor; NRTI—nucleoside reverse transcriptase inhibitor; NNRTI—non-nucleoside reverse transcriptase inhibitor; NVP—Nevirapine; PI—protease inhibitor; TDF—Tenofovir; VF—virological failure (defined as viral load ≥ 1000 copies/mL); 3TC—Lamivudine. ^1^ For participants without a DR test, a proxy test date was assigned based on the median time from enrollment to DR test among those who were tested. ^2^ Percentages may sum to more than 100 percent because participants can have drug resistance (DR) to multiple ART drug classes simultaneously.

**Table 2 viruses-17-01246-t002:** Factors associated with a drug resistance penalty score (DR-PS) < 30 (i.e., ART-suppressed or DR-PS < 30) versus a DR-PS ≥ 30 to any antiretroviral agent assessed by Stanford HIVdb that was detected in children enrolled in Opt4Kids studies (*n* = 704).

	Unadjusted RR ^1^	*p*-Value	Adjusted RR ^1^	*p*-Value
Child Clinical Characteristics				
Age category (years)				
1–5	2.66 (1.69, 4.18)	<0.001	1.84 (1.07, 3.14)	0.026
6–10	0.93 (0.60, 1.45)	0.753	0.77 (0.51, 1.16)	0.211
11–15 (Ref)	1.0			
Sex				
Male (Ref)	1.0			
Female	1.02 (0.70, 1.5)	0.901		
History of VF within two years prior to enrolling in Opt4Kids				
Yes	7.13 (4.72, 10.76)	<0.001	4.18 (2.77, 6.31)	<0.001
No (Ref)	1.0			
Base drug in ART regimen at time of DR test analysis				
NNRTI-containing (Ref)	1.0			
PI-containing	2.87 (1.73, 4.74)	<0.001	6.05 (3.43, 10.68)	<0.001
INSTI-containing	9.24 (5.61, 15.22)	<0.001	1.83 (1.08, 3.11)	0.026
Only NRTI-containing ^2^	-	-		
Co-administered NRTIs at time of DR test analysis ^3^				
ABC + 3TC (Ref)	1.0			
AZT + 3TC	1.31 (0.82, 2.12)	0.260		
TDF + 3TC	1.87 (1.17, 3.0)	0.009		
Caregiver Clinical Characteristics				
Age of caregiver				
<24 years	2.44 (1.25, 4.75)	0.009	1.10 (0.64, 1.87)	0.738
24–35 years	1.77 (1.17, 2.68)	0.007	1.28 (0.86, 1.91)	0.226
>35 years (Ref)	1.0			
Caregiver viral suppression				
Yes (Ref)	1.0			
No	1.91 (1.02, 3.58)	0.043	1.37 (0.74, 2.54)	0.321
Not applicable	1.17 (0.72, 1.90)	0.537	1.21 (0.80, 1.83)	0.356
Unknown	1.09 (0.62, 1.92)	0.769	1.42 (0.83, 2.45)	0.202
Child Psychosocial Characteristics				
School level				
Nursery (Ref)	1.0			
Primary	0.83 (0.50, 1.37)	0.458		
Secondary ^2^	-			
School type		0.832		
Day school	0.82 (0.13, 5.1)			
Boarding ^2^	-			
Mix of day and boarding (Ref)	1.0			
ART adherence				
Good (Ref)	1.0			
Poor	2.2 (1.51, 3.22)	<0.001	1.91 (1.32, 2.76)	0.001
Adolescence awareness of status				
No	1.1 (0.75, 1.62)			
Yes (Ref)	1.0	0.636		
Caregiver Psychosocial Characteristics				
Caregiver marital status		0.419		
Married	1.18 (0.79, 1.77)			
Unmarried (Ref)	1.0			
Unknown ^2^	-			
Caregiver educational attainment				
No Education	0.52 (0.13, 2.03)	0.345		
Primary	1.02 (0.69, 1.5)	0.937		
Secondary & above (Ref)	1.0			
Caregiver depression		0.122	1.02 (0.70, 1.49)	0.916
Yes	1.35 (0.92, 1.98)			
No (Ref)	1.0			
Caregiver stigma		0.543		
Yes	0.84 (0.47, 1.49)			
No (Ref)	1.0			
Caregiver HIV literacy		0.951		
Yes	0.99 (0.68, 1.44)			
No (Ref)	1.0			
Caregiver medicine administration confidence				
Yes (Ref)	1.0			
No	2.05 (1.16, 3.64)	0.014	1.89 (1.11, 3.22)	0.019
Caregiver intimate partner violence				
Yes	1.09 (0.7, 1.69)	0.704		
No (Ref)	1.0			
Household Psychosocial Characteristics				
Household food insecurity				
No (Ref)	1.0			
Yes	2.14 (0.32, 14.43)	0.434		
Other children in the household		0.382		
Yes	1.38 (0.67, 2.87)			
No (Ref)	1.0			
Other CLHIV in the household				
Yes (Ref)	1.0			
No	1.33 (0.73, 2.41)	0.349		
Structural Characteristics				
Time to facility				
<30 min (Ref)	1.0			
30 min to 1 h	0.54 (0.35, 0.83)	0.004	0.72 (0.47, 1.12)	0.144
>1 h	0.62 (0.38, 1.03)	0.064	0.71 (0.44, 1.14)	0.156
Clinic location				
Rural	0.81 (0.46, 1.44)	0.472		
Semi-urban	1.28 (0.83, 1.97)	0.257		
Urban (Ref)	1.0			
Clinic volume				
Light (Ref)	1.0			
Medium	0.56 (0.34, 0.95)	0.024	0.55 (0.33, 0.92)	0.023
Heavy	0.5 (0.28, 0.91)	0.032	0.76 (0.41, 1.39)	0.371

Abbreviations: ABC—Abacavir; ART—antiretroviral therapy; AZT—Zidovudine; CLHIV—children living with human immunodeficiency virus; DR-PS—drug resistance penalty score; NRTI—nucleoside reverse transcriptase inhibitor; NNRTI—non-nucleoside reverse transcriptase inhibitor; INSTI—integrase strand transfer inhibitor; IPV—intimate partner violence; PI—protease inhibitor; RR—relative risk; TDF—Tenofovir; VF—virological failure (defined as viral load ≥ 1000 copies/mL); 3TC—Lamivudine. ^1^ Relative risks were estimated using a Poisson regression model with robust standard errors; adjusted models included age, history of virological failure within two years prior to study, base drug in ART regimen at time of DR test analysis, age of caregiver, caregiver viral suppression, adherence, caregiver depression, medicine administration confidence, time to facility, and clinic volume. The observations with missing data were excluded; the extent of missingness is summarized in Table 1. ^2^ The relative risk cannot be calculated when the frequency of one of the categories is 0 which leads to quasi-complete separation and prevents proper estimation. ^3^ Not included in adjusted models due to collinearity with base ART regimen at index DRM result. Bold text indicates estimates with *p*-values < 0.05.

## Data Availability

The de-identified data, data codebook, and other data elements outlined in this study are available upon request from the corresponding author, ethics approval, submission of relevant documents, and signed data access agreement.

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
