# Peer review of "Clinical, Psychosocial, and Structural Factors Associated with the Detection of HIV Drug Resistance in Children Living with HIV in Kisumu, Kenya: Secondary Analysis of Data from the Opt4Kids Study"

_viruses, 2025, doi:10.3390/v17091246_

Round 1
Reviewer 1 Report
Comments and Suggestions for Authors
Determinants of DRM are always important to be found particularly in CLHIW. This research is well conducted and results are relevant in the field.
There are 3 points, which need clarification.
One is substantial, at page 5, line 195: "penalty score <30" = at least for the general description, I suggest to keep these participants as a separate group; I understand that the number may be quite low, but in my view it's important to maintain the distinction between <30 and >30.
Two are minor:
page 4, line 182: add the used version of Stanford Db;
page 15, line 374: add "Adjusted" in the X-axis legend.
Author Response
REVIEWER 1 COMMENTS:
Determinants of DRM are always important to be found particularly in CLHIW. This research is well conducted and results are relevant in the field. There are 3 points, which need clarification. One is substantial, at page 5, line 195: "penalty score <30" = at least for the general description, I suggest to keep these participants as a separate group; I understand that the number may be quite low, but in my view it's important to maintain the distinction between <30 and >30.
Response: We appreciate the reviewer's comment and have removed the terminology of “participants without any drug resistance testing” which referred to participants who did not undergo any DR testing and participants with DR penalty score (DR-PS) <30. We have also changed this wording in Tables 1 and 2, editing column labels to state “Children without DR testing or DR-PS<30” from the prior “Children without any drug resistance testing”. Although we changed the general description, we still group participants with DR-PS <30 (n=20) with participants who did not undergo any DR testing (i.e., did not experience viremia or high enough to undergo DR testing, n=591) in our analysis for the following reasons. Largely, we feel that participants with viremia but who have DR-PS<30 may not necessarily warrant a regimen change, which is the most consequential clinical care implication of undergoing a DR test. Given that few participants had a DR-PS<30, we felt that if we were to re-do our analysis to accommodate three separate analytical groups (i.e.,no DR testing, DR-PS<30, and DR-PS>30), interpreting a unit change in this new categorical variable would be ever more challenging. We believe leaving this variable as binary helps aid interpretations more easily. We also considered including a sensitivity analysis that included any DR test regardless of result (i.e., DR-PS>0), however came to the conclusion that this may also not provide clinically relevant information, given that DR-PS of 0-30 to a certain agent may not warrant a change in regimen or confer significant resistance. Of course, we had already included a sensitivity analysis that dropped the DR-PS to >15 (as well as >60) to toggle the various potentially clinically relevant PS cutoffs.
Two are minor:
page 4, line 182: add the used version of Stanford Db;
Response: Thank you for this important point. We had added the version of Stanford HIVdb used, to state “as assessed by the Stanford HIV drug resistance database (HIVdb)32 version 9.8 (2025-01-05) accessed on 21 April 2025”.
page 15, line 374: add "Adjusted" in the X-axis legend.
Response: We added “Adjusted” into the X-axis legend.
Reviewer 2 Report
Comments and Suggestions for Authors
The authors present a valuable study exploring the association between individual factors and the likelihood of developing potentially significant HIV drug resistance mutations (DRMs) among infected children in a high-burden region of Kenya, Kisumu County, during 2019–2023. The threshold for potential significance was defined as a score of 30 according to Stanford’s Genotypic Resistance Interpretation Algorithm.
The study population included groups of children previously examined in another project, supplemented by patients from the same study receiving dolutegravir-based regimens. HIV genotyping was performed in patients with detectable viral load, following a predefined algorithm.
The factors analyzed encompassed both standard clinical and demographic variables (sex, age, treatment regimen, history of virologic failure) and a broader set of psychosocial and contextual indicators. These included child-level factors (awareness of HIV status, adherence, etc.) and caregiver-level factors (family status, education, belief in the necessity of HIV treatment, stigma, etc.), as well as household-level characteristics (food insecurity, presence of other children in the household, including HIV-infected children) and selected structural characteristics (clinic location and volume).
The detailed statistical analysis revealed significant associations between DRMs and the following: child age of 1–5 years, history of virologic failure within the two years preceding the study, use of PI- or INSTI-containing regimens, poor adherence, and limited caregiver confidence in administering medication. Other variables examined did not demonstrate statistical significance.
While the study does not enable individual-level prediction of HIV drug resistance, it is an important contribution to epidemiological understanding and to the optimization of strategies aimed at preventing the emergence and spread of resistant strains in the population.
The manuscript would benefit from some clarifications and additional details, as outlined below:
- In the Methods section, specify exactly which treatment regimens were used in both the main Opt4Kids study and the expanded cohort, rather than listing them only in the Results and tables.
- Indicate the proportion of treatment failure for each regimen, with particular emphasis on dolutegravir-containing regimens (distinguishing between first- and second-generation integrase inhibitors).
- Figure 1: Provide a clear explanation for the term “unique participant.”
- Figure 2a: Consider removing individual mutation names, presenting only the drug classes.
- Clarify why no participants were found to have the typical NNRTI mutation K103N if possible.
- Specify the proportion of multiple resistance (i.e., several mutations to one or multiple drug classes) identified in the study.
Author Response
REVIEWER 2 COMMENTS:
The authors present a valuable study exploring the association between individual factors and the likelihood of developing potentially significant HIV drug resistance mutations (DRMs) among infected children in a high-burden region of Kenya, Kisumu County, during 2019–2023. The threshold for potential significance was defined as a score of 30 according to Stanford’s Genotypic Resistance Interpretation Algorithm. The study population included groups of children previously examined in another project, supplemented by patients from the same study receiving dolutegravir-based regimens. HIV genotyping was performed in patients with detectable viral load, following a predefined algorithm. The factors analyzed encompassed both standard clinical and demographic variables (sex, age, treatment regimen, history of virologic failure) and a broader set of psychosocial and contextual indicators. These included child-level factors (awareness of HIV status, adherence, etc.) and caregiver-level factors (family status, education, belief in the necessity of HIV treatment, stigma, etc.), as well as household-level characteristics (food insecurity, presence of other children in the household, including HIV-infected children) and selected structural characteristics (clinic location and volume). The detailed statistical analysis revealed significant associations between DRMs and the following: child age of 1–5 years, history of virologic failure within the two years preceding the study, use of PI- or INSTI-containing regimens, poor adherence, and limited caregiver confidence in administering medication. Other variables examined did not demonstrate statistical significance. While the study does not enable individual-level prediction of HIV drug resistance, it is an important contribution to epidemiological understanding and to the optimization of strategies aimed at preventing the emergence and spread of resistant strains in the population.
The manuscript would benefit from some clarifications and additional details, as outlined below:
1. In the Methods section, specify exactly which treatment regimens were used in both the main Opt4Kids study and the expanded cohort, rather than listing them only in the Results and tables.
Response: We appreciate the reviewer’s suggestion here and that specific ART regimens help better contextualize DRM development, and now have added in a new variable within Table 1, labeled “Full ART regimen at time of DR test analysis”. In the Results section, we have added the following sentence, “The most common full ART regimens at this time were abacavir/lamivudine/efavirenz, abacavir/lamivudine/lopinarvir/ritonavir, and zidovudine/lamivudine/lopinavir/ritonavir.”
2. Indicate the proportion of treatment failure for each regimen, with particular emphasis on dolutegravir-containing regimens (distinguishing between first- and second-generation integrase inhibitors)
Response: We appreciate the reviewer asking for more information about viral failure based on specific regimens, especially for those on dolutegravir-containing regimens. In fact, we have a full manuscript forthcoming on viral failure among this cohort as they transitioned to dolutegravir-containing regimens. Therefore, we definitely plan to dive into this component of the story, but we think it will be distracting from the main messaging here to include this component here. For now, because being able to conduct a DRM test in the first place is a proxy for viremia, at least viral load >1000 copies/mL via Sanger sequencing (and lower thresholds for other sequencing platforms), in many ways that information is parallel to the various ART regimens data now included in Table 1, including the new variable now included based on this reviewer’s comment #1.
3. Figure 1: Provide a clear explanation for the term “unique participant.”
Response: We agree and have changed terminology from “Unique participants” to
“Non-overlapping participants.”
4. Figure 2a: Consider removing individual mutation names, presenting only the drug classes.
Response: We really appreciate this point that the individual mutation names can be overwhelming, especially for non-virologists or -clinicians who are not intimately familiar with these. Given that there are so many clinical nuances to which mutations exist, sometimes even in combination with others (which is a level of complexity that our current analysis avoids for this very reason), we have chosen to keep the individual mutation names in Figure 2. However, we are spacing them out on Figure 2a so that the layout is more reader friendly.
5. Clarify why no participants were found to have the typical NNRTI mutation K103N if possible.
Response: We are so grateful for the reviewer’s close examination of our figures, noting that a common K103N mutation was missing, as it prompted us to examine our code more carefully. This revealed a data coding error on the donut figures from our end. We have edited the donut figures to accurately characterize mutations across the 93 participants. The NNRTI mutation of K103N has been identified and corrected for in our figures. We have triple checked our code now to ensure accuracy of findings.
6. Specify the proportion of multiple resistance (i.e., several mutations to one or multiple drug classes) identified in the study.
Response: Thank you for this comment. We have edited the manuscript according to this comment. In Table 1, we have now added a variable, “Frequency of resistance to multiple drugs (DR-PS >30) within an ART drug class” to highlight these details. In our Results section, we have also added the sentence, “Most participants were resistant to multiple ARVs within a drug class. 82 (88.2%) participants had a DR-PS≥30 to two or more NNRTIs, and 69 (74.2%) had a DR-PS≥30 to two or more NRTIs (Table 1)”.